# Salivary Interleukin-13 and Transforming Growth Factor Beta as Potential Biomarkers of Cancer Cachexia

**DOI:** 10.3390/cancers16173035

**Published:** 2024-08-30

**Authors:** Borislav Belev, Ivan Vičić, Filip Sedlić, Matko Prtorić, Majana Soče, Juraj Prejac, Slavica Potočki, Tajana Silovski, Davorin Herceg, Ana Kulić

**Affiliations:** 1Department of Oncology, University Hospital Center Zagreb, 10000 Zagreb, Croatia; ivan.vicic1@gmail.com (I.V.); majana.soce@gmail.com (M.S.); juraj.prejac@gmail.com (J.P.); tsilovski@gmail.com (T.S.); davorinh1@gmail.com (D.H.); 2School of Medicine, University of Zagreb, 10000 Zagreb, Croatia; filip.sedlic@mef.hr (F.S.); matko.prtoric35@gmail.com (M.P.); 3School of Dental Medicine, University of Zagreb, 10000 Zagreb, Croatia; 4Department of Medical Chemistry, Biochemistry and Clinical Chemistry, School of Medicine, University of Zagreb, 10000 Zagreb, Croatia; spotocki@mef.hr

**Keywords:** saliva, cancer cachexia, cancer metabolism, biomarker, cytokines, TGF-ß, IL-13

## Abstract

**Simple Summary:**

Cancer cachexia is a complex metabolic condition that is often overlooked and recognized in a late irreversible phase. There is a continuing effort to discover and define a biochemical biomarker of the condition. In this study, we have chosen interleukin-13 (IL-13) and transforming growth factor beta (TGF-β), i.e., two cytokines with presumed roles in the development of cancer cachexia, and we measured their concentrations in the serum and saliva of cachectic patients with metastatic solid tumors, non-cachectic patients with metastatic solid tumors, and healthy individuals. We have demonstrated the role of saliva cytokine measurement as a potential sample for cachexia investigations as opposed to the standard approach to biomarker research, which is serum measurement. We have also found that the salivary IL-13 and TGF-ß are independent risk factors and thus could serve as potential biomarkers of the condition, a fact that warrants further research and confirmation.

**Abstract:**

Cancer cachexia is a syndrome characterized by weight and muscle loss and functional impairment, strongly influencing survival in cancer patients. In this study, we aimed to establish the role of saliva cytokine measurement in cancer cachexia investigation and define two potential independent salivary biomarkers of the condition. Methods: serum and saliva specimens were obtained from 78 patients. Forty-six patients were non-cachectic, and 32 patients were cachectic (per SCRINIO group criteria), all with metastatic solid tumors. Commercial ELISA kits were used to determine the salivary and serum concentrations of interleukin 13 (IL-13) and transforming growth factor beta (TGF-β) in two patient groups and healthy controls. Laboratory values were obtained from the hospital information system, and weight and height were measured at the time of sampling. Results: A statistically significant difference was observed between the groups in saliva IL-13 concentrations but no difference in serum concentrations. Statistically significant differences were also observed between the groups in saliva and serum concentrations of TGF-β. Logistic regression analysis has identified salivary IL-13 and TGF-β as independent factors for cancer cachexia. Conclusions: We demonstrated saliva as a valuable specimen for cachexia investigation and established IL-13 and TGF-β as potential cancer cachexia biomarkers. Further research is needed to evaluate these findings.

## 1. Introduction

Cancer cachexia presents as a multiorgan syndrome marked by the depletion of weight and muscle mass, also including loss of fat tissue, which has a severe impact on survival and quality of life in cancer patients [1]. Cancer cachexia advances from pre-cachexia to cachexia, and, typically, in fully-developed cachexia, established nutritional support is insufficient for complete reversal. At the same time, chronic disease in the background, like cancer, sustains this progression of metabolic deterioration, leading to poor outcomes at the end [2]. Although differently present in various tumors, it becomes an important consequence of tumor growth and progression. Cachexia is observed in 80% of gastric, pancreatic, and esophageal cancer patients, 70% of individuals with head and neck tumors, and 60% of colorectal and prostate cancer [3,4]. It leads to death in at least 22% of all cancer patients. Therefore, it is of great importance to recognize cachexia as soon as possible because of the need for early intervention [5].

Many cytokines are proposed as potential biomarkers of early cachexia, but data still need to be uniquely confirmed regarding their clinical and pathophysiological role in this process [6]. Generally, the majority of identified potential biomarkers are pro-inflammatory cytokines, like interleukin-6 (IL-6) and tumor necrosis factor-alpha (TNF-α), or fat and muscle metabolism-related molecules (free fatty acids, glycerol, β-dystroglycan, sphingolipids, hexosyl- and lactosyl-ceramides) [7].

Transforming growth factor beta (TGF-β) has been previously identified as a mediator of cancer progression, and more recently, a role in muscle wasting has emerged [8]. Through canonical TGF-β signaling, ryanodine receptor oxidation and nitrosylation are mediated, which impairs muscle contraction [9]. The effect of TGF-β has also been observed in the extracellular matrix through a process of epithelial-to-mesenchymal transition (Figure 1a), but it is also known for its indirect action on immune homeostasis [10,11,12]. Generally, TGF-β is important both for various metabolic signaling and metabolic reprogramming [13].

Interleukin-13 (IL-13), on the other side, is an important muscle metabolism mediator previously described as a facilitator of mitochondrial biogenesis and fatty acid oxidation (Figure 1b). Through signal transducer and activator of transcription 3 (STAT3) and possibly alternative intracellular pathways, this cytokine has been implied to participate in muscle metabolism adaptation in response to increased stress [14]. IL-13 has also been implicated in cancer invasion and progression by paracrine and autocrine signaling in the tumor microenvironment [15,16]. The pleiotropic nature of IL-13 signaling encompasses muscle metabolism and metastatic tumor burden [15]. This fact encouraged us to consider its role as a possible cancer cachexia mediator, which has not been established so far.

Although saliva is not a standardized source of biomarkers for the cachexia condition, this study aimed to determine if it is an appropriate sampling source for cancer cachexia evaluation by comparing the serum and salivary concentrations of two cytokines presumed to have a role in cachexia pathogenesis and possibly establishing them as potential biomarkers of the condition.

## 2. Materials and Methods

### 2.1. Patient Selection and Sampling

This cross-sectional observational study was conducted from February 2017 to June 2023 at the University Hospital Center Zagreb, Department of Oncology. A total of 78 patients were recruited during the treatment and follow-up. Their leading physician recognized the patients as cachectic and reported them to the study team. A healthy control group consisted of 20 volunteers. The majority of patients had gastrointestinal cancers (71%), followed by breast cancer (13%), and other cancer types (prostate, soft tissue sarcoma, urothelial cancer). Patients were divided into two groups according to the presence of cancer cachexia, which was defined per the criteria of the SCRINIO working group and the body mass index (BMI) of less than 20 kg/m^2^ [17,18]. Cachectic patients were those who lost 10% of their body weight and had at least one of the symptoms of early satiation, anorexia, and fatigue and had a BMI of less than 20 kg/m^2^. Non-cachectic patients (and healthy controls) had no body mass loss and a BMI higher than 20 kg/m^2^. All the patients had metastatic disease at the time of the sampling and were treated according to the specific cancer type (no pretreatment sampling was performed, sampling in all patients after at least one radiological evaluation, and no sampling during supportive care). The metastatic status was confirmed by the most recent radiological exams (reevaluation CT scans). There were no differences between the groups concerning age and gender. The exclusion criteria for all study participants were bowel obstruction (defined as the clinical suspicion that is radiologically confirmed), diagnosis of head and neck cancer, active infection or sepsis at the time of sample collection (clinical suspicion or radiological evidence or antibiotic, antifungal or antiviral therapy with curative intent), long-term (more than two weeks) oral corticosteroid use (more than 7.5 mg of oral prednisone or equivalent dose), liver failure (patients with liver cirrhosis previously established), and the presence of an autoimmune disease (Sjögren’s syndrome, systemic lupus erythematosus, systemic sclerosis, rheumathoid arthritis, axial spondyloarthritis, autoinflammatory disease, vasculitis, inflammatory bowel disease, autoimmune liver disease confirmed by a rheumatologist). The exclusion criteria were checked on the sampling day before the sampling procedure (for patients and healthy controls equally). The patients’ medical history data and laboratory results used in the study (aspartate aminotransferase (AST), alanine aminotransferase (ALT), C-reactive protein (CRP), total protein, and albumin concentrations) were obtained from the hospital information system and were measured as a part of the routine laboratory tests no more than 5 days before the sampling day. The sampling procedure consisted of two parts, conducted one after the other (all the samples were collected between 9 a.m. and 2 p.m.). The sampling procedure was conducted one day after recruitment. There were eighty-two patients eligible for the sampling procedure and fulfilled the criteria, but in four of them, we had no adequate quantity of sample (saliva). None of the patients withdrew their consent. The first part was blood sampling by venepuncture (collecting 3 mL of blood in a Vacutainer EDTA blood collecting tube). The second part was the saliva sampling. Three mL of whole unstimulated saliva was collected by pooling the saliva in the mouth of the subject for 60 s and spitting it into the plastic tube thereafter. This procedure was repeated until the designated volume was reached. The subjects were instructed to abstain from all food and drinks for at least an hour before the sampling and to rinse the mouth with clean water precisely one hour before the sampling. The saliva samples were inspected for macroscopic contamination and food residues, and all such participants were excluded from the study. Weight and height were measured on the day of the sampling.

### 2.2. Sample Processing

All the samples were processed within the first hour from sampling to minimize proteolysis of salivary proteins and, by that time, were stored at +4 °C. The blood samples were centrifuged for 10 min at 3000 rpm. After that, the serum was aliquoted into labeled cryovials and then stored in an ultra-freezer at −80 °C. After centrifugation at 4000 rpm for 20 min, the clarified saliva supernatant was aliquoted and collected. All the samples were stored at −80 °C for further use.

Enzyme-linked immunosorbent assay (ELISA) for serum and saliva IL-13 and TGF-β1 was performed using a commercially available ELISA kit (IL-13: ELISA Q311065 kit, eBioscience; TGF-β1: ELISA Lot 19599203 kit, Invitrogen, Waltham, MA, USA). The analysis was performed according to the manufacturer’s instructions. The results were calculated by constructing a standard curve.

### 2.3. Statistical Analysis

The results were analyzed using the statistical software MedCalc (v 22.030). The differences in the distribution of the numerical variables have been analyzed with the Mann–Whitney test. The distribution of IL-13 and TGF-β in the serum and saliva of patients with malignant disease and the control group was analyzed using the Kruskal-Wallis ANOVA test. Spearman rank correlation coefficients were used for correlation assessment. Multivariate analysis was performed using logistic regression analysis. A *p*-value < 0.05 was considered statistically significant. The diagnostic accuracy and the optimal cut-point value for the IL-13 and TGF-β in serum and saliva levels between the two groups have been obtained based on the value of the AUC, area under the ROC curves (receiver operating characteristic curve).

## 3. Results

Demographic and clinical characteristics of cancer patients are shown in Table 1. The median values of IL-13 and TGF-β in normal serum were 11.2 pg/mL (3.27–32.4 pg/mL) and 31 pg/mL (8.6–48.5 pg/mL). Median values of IL-13 in normal saliva were 28.6 pg/mL (12.3–66.6 pg/mL) and of TGF-β 40.5 pg/mL (10.2–66.6 pg/mL). The control group included thirteen men (65%) and seven women (35). The median range age of healthy controls was 53 (range 45–59 years). The levels of IL-13 and TGF-β in the patients’ serum and saliva are shown in Table 1.

Concentrations of IL-13 and TGF-β in the serum and saliva of patients with malignant diseases were statistically significantly higher than in the serum and saliva of healthy subjects (Figure 2a,b).

A positive correlation was found between IL-13 concentrations in serum and saliva (*p* < 0.001) as well as for TGF-β (*p* < 0.001) (Figure 3a,b).

Figure 4a shows the correlation of IL-13 and TGF-β in the saliva of patients (*p* < 0.001) and no correlation between the two parameters in the serum of patients (*p* = 0.112) (Figure 4b).

The levels of IL-13 in the serum did not show a difference between cachexia and non-cachexia groups (Figure 5a), while serum levels of TGF-β showed a difference between cachectic and non-cachectic patients (Figure 5b). Salivary levels of both cytokines were higher in patients with cachexia (Figure 5c,d).

The diagnostic accuracy of IL-13 and TGF-β in the serum and saliva of patients with and without cachexia was determined by ROC curve analysis. In our study, the AUC value for IL-13 in serum was 0.594 (sensitivity 75%, specificity 50%, cut-off value of 34.3 pg/mL), and for saliva, it was 0.820 (sensitivity 90.6%, specificity 63%, cut-off value 68.4 pg/mL) (Figure 6a,b). The AUC values for TGF-β in serum were 0.869 (sensitivity 81.2%, specificity 91.3%, cut-off value 113.3 pg/mL), and in saliva, 0.913 (sensitivity 81.2%, specificity 97.8%, cut-off value 157.4 pg/mL) (Figure 7a,b).

A statistically significant negative correlation was found between IL-13 in serum and saliva and the BMI of non-cachectic patients (*p* = 0.009; *p* = 0.008) (Table 2). The levels of TGF-β in the serum of cachectic and non-cachectic patients were negatively correlated with BMI (*p* = 0.045; *p* = 0.010) (Table 2), and the correlation was statistically significant. In patients with cachexia, a negative statistical association was found between salivary TGF-β and BMI (*p* = 0.045) (Table 2).

Logistic regression analysis showed that high saliva TGF-β and IL-13 levels were independent risk factors for cachexia (Table 3). Low serum albumin concentrations and BMI are also independent factors associated with cachexia (Table 3).

## 4. Discussion

Cachetic patients with malignant diseases have shorter survival, a reduced response to therapy, and a reduced quality of life [19]. It would be of great importance to recognize the early phase of developing cancer cachexia not only for the prognosis of the disease but for treatment strategies as well. It is very well known that early cancer cachexia, so-called precachexia, is usually overseen. Many cytokines, especially proinflammatory ones, and chemokines, as well as markers of fat loss (like leptin, glycerol, etc.), have certainly some role in this highly complex pathophysiological process [20]. Although serum has been almost exclusively used as an investigated sample, saliva is also a potential source of biomarkers in different biological processes that are still under investigation. Salivary diagnostics is an approach that offers a non-invasive, atraumatic, simple, and cost-effective way of detecting proteins, nucleic acids, and metabolites that are secreted by ultrafiltration or exosome-like microvesicles, as was previously shown in various cancer models and reflects systemic conditions [21,22].

In this study, we measured saliva and serum concentrations of TGF-β and IL-13 in cachectic and non-cachectic patients with metastatic cancer and healthy controls. IL-13 and TGF-β concentrations in the saliva of healthy controls were higher than those in the blood, which is consistent with some studies [23,24].

Significantly higher concentrations of TGF-β in the serum of cancer cachexia patients measured in our study are in accordance with previous research. The role of TGF-β has been established in adipose tissue structural changes in cancer cachexia, as well as a mediator of muscle atrophy [9,25,26]. However, previous studies have measured salivary TGF-β in various conditions like oral submucous fibrosis, oropharyngeal squamous cell carcinoma, and radiation-induced salivary gland damage, but to our best knowledge, there are no published data on salivary TGF-β concentrations in cancer cachexia without the primary disease involvement of head and neck structures [23,27,28]. Our study findings confirm the elevated serum concentration of TGF-β in metastatic cancer patients compared with healthy controls but also an elevated salivary concentration. Although the concentration is elevated in all metastatic patients compared with healthy controls, there is a significantly higher concentration in cancer cachexia patients, with a clear correlation between serum and saliva, which implies TGF-β could be considered a potential biomarker of cancer cachexia. ROC analysis shows high AUC values (0.913) for saliva and very good AUC values (0.869) for plasma TGF-β. These high AUC values for saliva and very high values for plasma suggest that TGF-β values could be a good potential biomarker of cancer cachexia in patients with malignant disease. The cause of the increased concentration of TGF-β in saliva compared with serum might be the relatively small molecular weight of TGF-β, which enables better filtration in saliva [29]. Another possible cause is that serum concentration oscillates more than saliva [24].

IL-13 concentrations have been previously measured in cancer patients. One study measured its salivary concentration in oral squamous cell carcinoma as a potential early detection and progression biomarker [30]. Furthermore, serum concentrations have been measured in colon cancer patients as a part of a prognostic tool for rapid progression and 5-FU-based therapy response [31]. To our knowledge, there are no available data on serum or saliva concentrations of IL-13 in cachectic patients. A higher concentration in metastatic patients compared with healthy controls is expected, as is a positive correlation between salivary and serum concentrations. Although serum concentrations of IL-13 did not differ in cancer cachexia compared with non-cachectic cancer patients, we have measured significantly higher concentrations of salivary IL-13 in cachectic patients, which could imply a salivary IL-13 as a possible biomarker of cancer cachexia and possibly establish the role of salivary cytokine measurement in the advanced cancer setting as a more reliable and informative parameter, what was not known so far. The very high AUC (0.812) of saliva IL-13 confirms the aforementioned results.

We found a statistically significant correlation between salivary concentrations of IL-13 and TGF-β1, which could imply part of the common pathogenesis of the condition. Using logistic regression analysis, we found salivary TGF-β and IL-13 levels as independent factors of cancer cachexia, while serum concentrations are not an independent factor. Logistic regression analysis also identifies albumin as an independent factor of cachexia. In our study, gender, age, CRP, AST, ALT, and total protein were not shown to be independent factors of cancer cachexia.

Both examples support using saliva as a sample that offers insight into the systemic metabolic condition of cancer cachexia and provides new information about the concentration of potential biomarkers TGF-β and IL-13. Additionally, diagnostic tools for assessing muscle strength and body composition are still not widely used due to non-standardized methods and uneven availability [32,33]. Although many potential candidates have been proposed as biomarkers for metabolic changes in cachexia, none have been clinically established as such [7]. Further research is needed to investigate potential biomarkers of cancer cachexia in saliva, but in this study, we demonstrated a possible advantage over serum measurements.

Limitations of this study include the lack of sarcopenia measurement, an important component of cancer cachexia. In this study, cachexia was defined by the SCRINIO group criteria (weight loss of 10% and presence of one of the symptoms of early satiation, anorexia, or fatigue, as well as the restriction of BMI of less than 20 kg/m^2^, as it was previously defined by ESMO guidelines) [17,18]. In that way, we avoided specific differences in phenotype that might unfairly exclude some of the apparently cachectic patients. The groups were not matched by tumor sites, which might have skewed the results, as well as the different specific cancer therapies, which by themselves could affect muscle and fat metabolism. We included different cancer types in this analysis, assuming they share a common cachexia pathogenic mechanism [34]. The age of patients was not limited to a specific age group, which might have influenced the saliva composition and potentially altered the results. The exclusion criteria were chosen in an attempt to minimize high cytokine production in an obvious inflammatory condition such as infection, autoimmune, or autoinflammatory disease, which is an imperfect process, especially in metastatic cancer conditions. We excluded head and neck patients since elevated cytokine levels were observed in this population of patients [23]. We also excluded patients with a bowel obstruction in order to exclude mechanical causes of low-calorie intake.

## 5. Conclusions

In this analysis, we demonstrated saliva as a valuable specimen for cachexia investigation and established IL-13 and TGF-β as potential cancer cachexia biomarkers. This finding requires additional confirmation in similar research and offers new insight into the complex cachexia pathophysiology, potentially relevant for clinical practice.

## Figures and Tables

**Figure 1 cancers-16-03035-f001:**
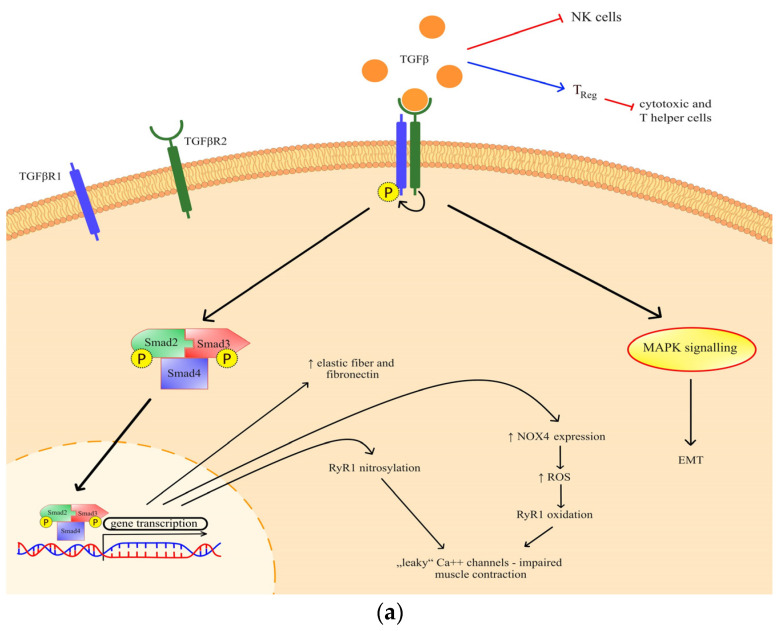
(**a**) The effects and signaling of TGF-β. Canonical signaling results in a change of gene transcription, which exerts specific effects; (**b**) The effects and signaling of IL-13. Through STAT3 signaling and possibly another, so far undefined mechanism, IL-13 exhibits its metabolic effects. TGF-βR—transforming growth factor beta receptor; RyR—ryanodine receptor; NOX 4—NADPH oxidase 4; ROS—reactive oxygen species; MAPK—mitogen-activated protein kinase; EMT—epithelial-mesenchymal transition; Err—estrogen-related receptor; IL-4R—interleukin-4 receptor; IL-13R—interleukin-13 receptor; STAT 3—signal transducer and activator of transcription 3.

**Figure 2 cancers-16-03035-f002:**
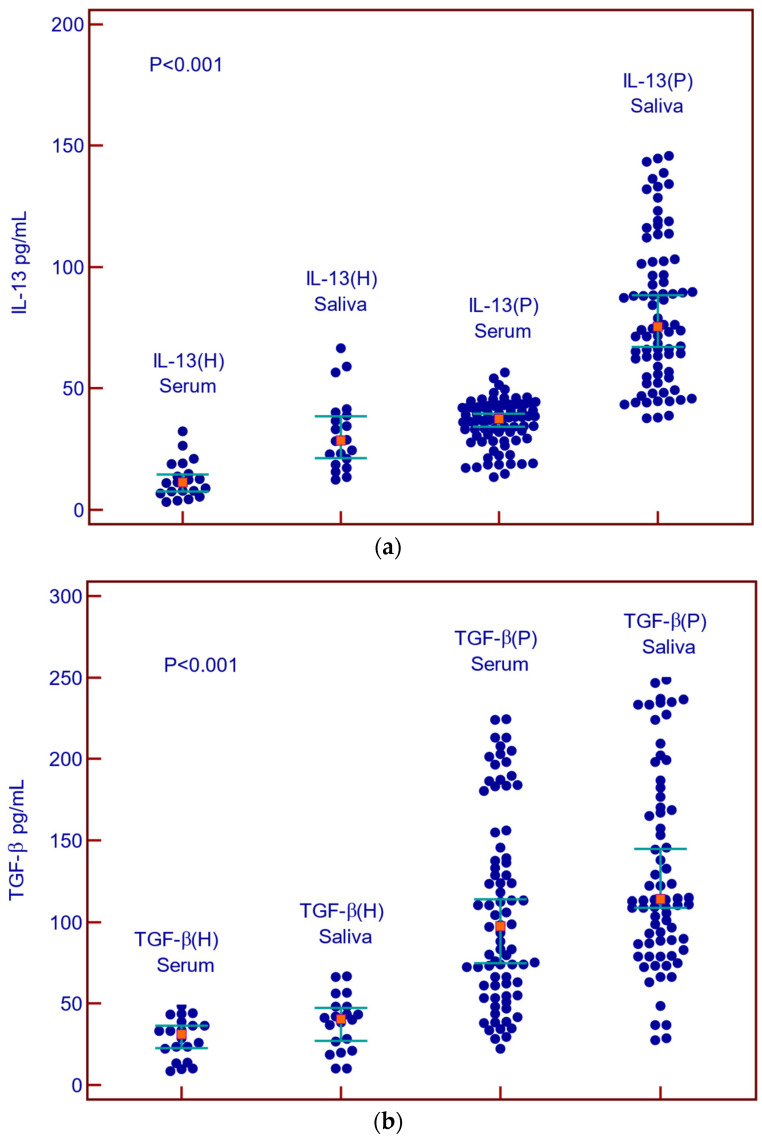
Concentrations of IL-13 and TGF-β in serum and saliva of patients with malignant diseases and healthy controls. (**a**) Summary data of serum and saliva IL-13 concentrations in healthy control (H) and malignant diseases (P); (**b**) Summary data of serum and saliva TGF-β in healthy control (H) and malignant diseases.

**Figure 3 cancers-16-03035-f003:**
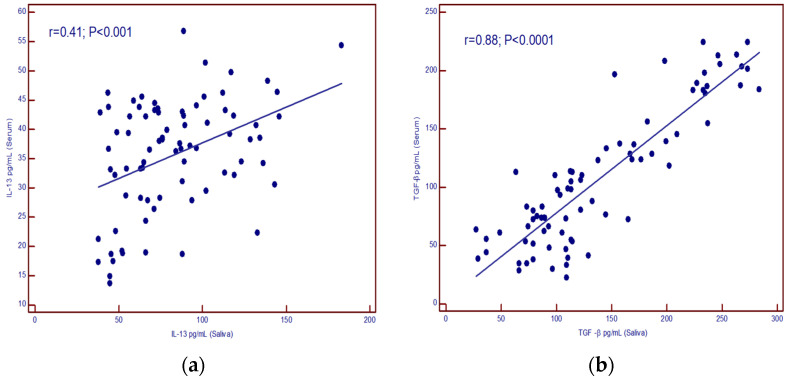
(**a**) correlation of IL-13 levels in serum and saliva; (**b**) correlation of TGF-β levels in serum and saliva.

**Figure 4 cancers-16-03035-f004:**
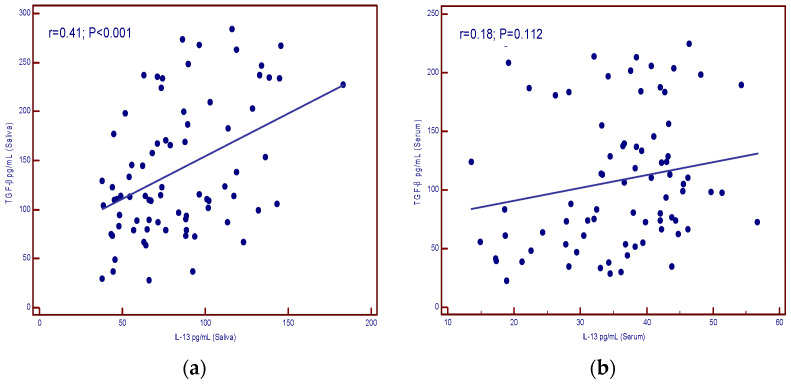
Correlation of IL-13 and TGF-β levels in patients with malignant diseases. (**a**) saliva (significant correlation); (**b**) serum (no correlation).

**Figure 5 cancers-16-03035-f005:**
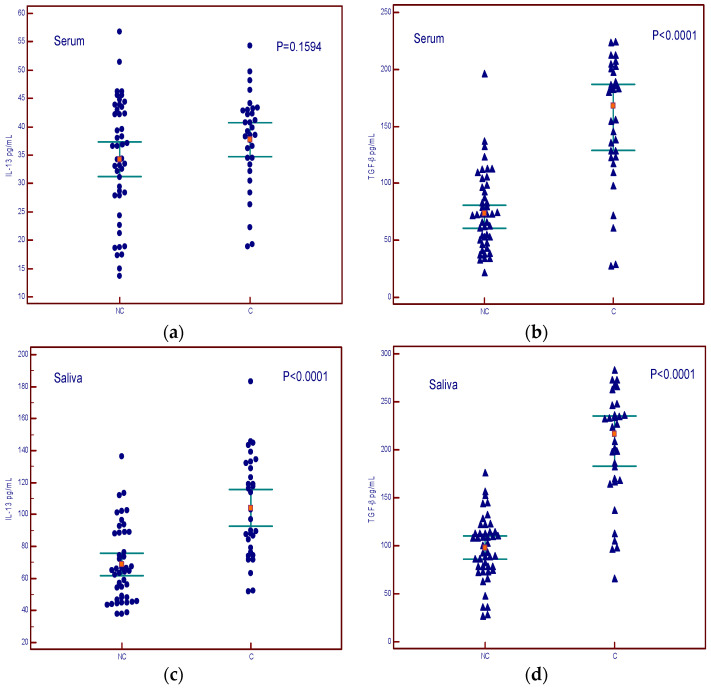
Concentrations of IL-13 and TGF-β in serum and saliva of non-cachectic (NC) and cachectic patients (C). (**a**) summary data of serum IL-13 concentrations; (**b**) summary data of serum TGF-β concentrations; (**c**) summary data of saliva IL-13 concentrations; (**d**) summary data of saliva TGF-β concentrations.

**Figure 6 cancers-16-03035-f006:**
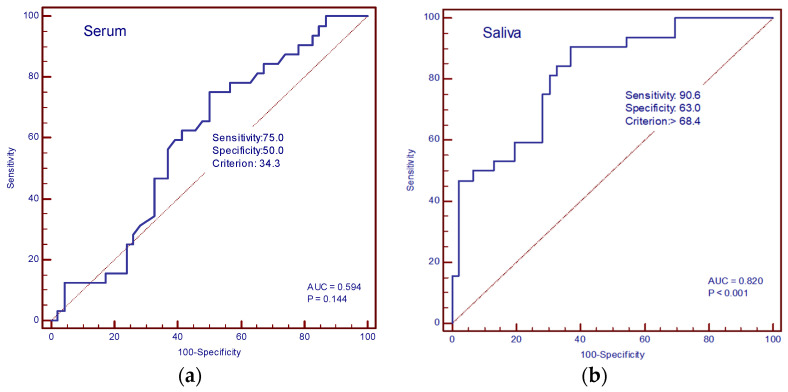
ROC curve analysis. (**a**) AUC of serum IL-13; (**b**) AUC of saliva IL-13.

**Figure 7 cancers-16-03035-f007:**
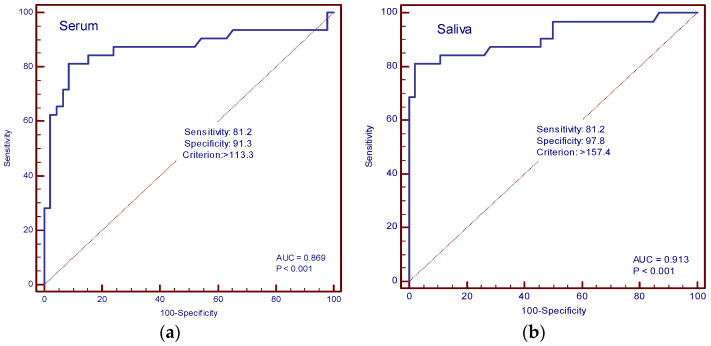
ROC curve analysis. (**a**) AUC of serum TGF-β; (**b**) AUC of saliva TGF-β.

**Table 1 cancers-16-03035-t001:** General characteristics of cancer patients with laboratory test results and measured cytokine levels.

				Cachexia		
			N = 78	NC, N = 46	C, N = 32	*p*
Gender	F	N	34	20	14	0.231
		%	44	59	41	
	M	N	44	26	18	
		%	56	59	41	
Age (y)		median	52	60	60	0.170
		range	48–62	21–78	37–74	
BMI (kg/m^2^)		median	23.4	26.6	17.74	**<0.001**
		range	15.5–40.4	20–40.4	15.4–19.8	
CRP (mg/mL)		median	6.7	7.6	3.2	0.100
		range	0.5–112	1–112	0.5–64.7	
AST (U/L)		median	25	25	28.5	0.783
(RV 11–38 U/L)		range	8–115	9–115	8–83	
ALT (U/L)		median	18	19	17	0.683
(RV 12–48 U/L)		range	6–146	6–146	6–86	
Albumin (g/L)		median	37.2	38.2	37.1	0.041
(RV 40.6–51.4 g/L)		range	22.3–47.8	28.3–47.8	22.3–43.1	
Total protein (g/L)		median	69.5	71	68	0.226
(RV 66–81 g/L)		range	52–91	54–91	52–77	
IL-13 (pg/mL)		median	37.4	35.4	38.5	0.159
(serum)		range	13.6–56.7	13.6–56.7	18.8–54.3	
IL-13 (pg/mL)		median	75.4	64.2	99.9	**<0.001**
(saliva)		range	37.8–183.3	37.8–136.4	52.1–183.3	
TGF-β (pg/mL)		Median	97.6	73.5	168.3	**<0.001**
(serum)		Range	22.1–224.5	22.1–196.3	28.3–224.5	
TGF-β (pg/mL)		Median	114	97.4	216.6	**<0.001**
(saliva)		range	27.3–283.5	27.3–176.5	66.3–283.5	

NC: non-cachectic; C: cachectic; BMI: body mass index; CRP: C-reactive protein; AST: aspartate aminotransferase; ALT: alanine aminotransferase; RV: reference value; IL-13: interleukin 13; TGF-β: transforming growth factor beta.

**Table 2 cancers-16-03035-t002:** Correlation between IL-13 and TGF-β in saliva and serum of cachectic (C) and non-cachectic (NC) patients and clinical and biochemical parameters.

	IL-13 (Serum)		IL-13 (Saliva)		TGF-β (Serum)		TGF-β (Saliva)	
	NC	C	NC	C	NC	C	NC	C
Gender	0.734	0.516	0.643	0.551	0.510	0.713	0.100	0.137
Age	0.680	0.354	0.117	0.855	0.662	0.669	0.721	0.358
BMI	0.009	0.076	0.008	0.061	0.010	0.045	0.125	0.045
CRP	0.741	0.546	0.208	0.647	0.370	0.360	0.482	0.453
AST	0.560	0.894	0.566	0.120	0.424	0.480	0.373	0.371
ALT	0.718	0.810	0.140	0.086	0.651	0.895	0.721	0.358
Albumin	0.404	0.622	0.068	0.287	0.149	0.297	0.489	0.672
Total protein	0.617	0.196	0.771	0.607	0.411	0.468	0.100	0.137

NC: non-cachectic; C: cachectic; BMI: body mass index; CRP: C-reactive protein; AST: aspartate aminotransferase; ALT: alanine aminotransferase; IL-13: interleukin 13; TGF-β: transforming growth factor beta.

**Table 3 cancers-16-03035-t003:** Logistic regression analysis comparing cachexia versus without cachexia patients. B—partial regression coefficient; SE—standard error; β—standard partial regression coefficient.

Variables	B	SE	β	*p*
Constant	1.62			
Gender	0.129	0.073	0.107	0.085
Age	−0.003	0.003	−0.151	0.311
BMI	−0.023	0.009	−0.370	0.010
CRP	−0.002	0.001	−0.193	0.197
AST	0.001	0.001	0.075	0.611
ALT	−0.002	0.001	−0.190	0.199
Albumin	−0.012	0.005	−0.322	0.027
Total protein	0.003	0.005	0.081	0.556
IL-13 (serum)	−0.045	0.071	−0.093	0.528
IL-13 (saliva)	0.173	0.082	0.295	0.020
TGF-β (serum)	−0.226	0.141	−0.231	0.113
TGF-β (saliva)	0.591	0.154	0.491	0.001

B: unstandardized beta; SE: the standard error of the unstandardized beta; β: standardized beta; BMI: body mass index; CRP: C-reactive protein; AST: aspartate aminotransferase; ALT: alanine aminotransferase; IL-13: interleukin 13; TGF-β: transforming growth factor beta.

## Data Availability

The data presented in this study are available upon request from the corresponding author.

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
