# Peer review of "Salivary Interleukin-13 and Transforming Growth Factor Beta as Potential Biomarkers of Cancer Cachexia"

_cancers, 2024, doi:10.3390/cancers16173035_

Round 1

Reviewer 1 Report

Comments and Suggestions for Authors

Line 20, 40, 72 should be reviewed for grammatical errors.

Figure 1 is poorly constructed and requires a more detailed figure legend. This figure should also depict a nucleus. The interaction of Smad 2 and 3 with 4 should result in their relocation to the nucleus where they impact transcription. However, this reaction seems to occur in the cytosol. A figure legend would clear this up.

Figure 2b the boxes should be removed and replaced with a β symbol.

Figure 4a legend should state that no correlation was found in the serum or saliva of patients.

Figure 5 should be labeled with IL-13 or TGF-β.

The acronym ROC should be described and defined.

Was the cachexia staged to determine if the concentration of these cytokines increase as cachexia progresses? Or, to determine if saliva can be used as an early marker in the diagnosis of cancer.

Also, information should be provided on whether the disease was metastatic which could also play a role in concentration of these markers that seem to also be markers of a cancer diagnosis.

The correlations are a bit confusing. The authors state that they found a correlation between TGF-β and IL-13, however, I don’t see that depicted in a graph. Furthermore, it’s very difficult to understand why BMI and cachexia are not correlated in this study.

Comments on the Quality of English Language

Small grammatical errors. Overall the grammar is good. 

Author Response

Thank you very much for taking time to review this manuscript. Please, the detailed responses on your questions/suggestions/comments, in the text, highlighted in green

Comments 1. Line 20, 40, 72 should be reviewed for grammatical errors.

Response 1. Thank you for your suggestions - lines 20, 40, and 72 were reviewed for grammatical errors.

Comments2. Figure 1 is poorly constructed and requires a more detailed figure legend. This figure should also depict a nucleus. The interaction of Smad 2 and 3 with 4 should result in their relocation to the nucleus where they impact transcription. However, this reaction seems to occur in the cytosol. A figure legend would clear this up.

Response 2. We find your comment very useful- Figure 1 is corrected. The nucleus is added, and a legend is expanded.

Comments 3. Figure 2b, the boxes should be removed and replaced with a β symbol.

Response 3. Thank you, we made correction as instructed

Comments 4. Figure 4a legend should state that no correlation was found in the serum or saliva of patients.

Response 4 : This is very important question, thank you for this- correction is made: the legend of Figure 4a states that no correlation was found in the serum, but a significant correlation was found in the saliva of patients (line 224)

Comments 5. Figure 5 should be labeled with IL-13 or TGF-β.

Response 5. Figure 5 Individual figures are labeled with IL-13 or TGF-β on the ordinate axis.

Comments 6. The acronym ROC should be described and defined.

Response 6. The acronym ROC is described and defined (line169)

Comments/Question 7. Was the cachexia staged to determine if the concentration of these cytokines increase as cachexia progresses? Or, to determine if saliva can be used as an early marker in the diagnosis of cancer.

Response 7. Very good question, thank you. The cachexia was not staged, and the study design did not aim to answer whether these cytokines levels increase as the disease progresses – this study only evaluated the levels in patients with metastatic cancer vs metastatic cancer without cachexia – the question of these cytokine use is to be determined in future research.

Comments 8. Also, information should be provided on whether the disease was metastatic which could also play a role in concentration of these markers that seem to also be markers of a cancer diagnosis.

Response 8.  All the patients were metastatic at the time of recruitment and sample collection (line115-119). The distinction was made in order to describe the difference that cancer cachexia produces, i.e., all the patients had metastatic disease, but the difference was the presence of cachexia (healthy individuals, of course, had no cancer). Also, all the patients received cancer therapy at the time of the sample collection. It is also stated in the limitations of the study that the patient groups were not matched by the specific cancer types and therapy types, which could have an impact on the results.

Comments 9. The correlations are a bit confusing. The authors state that they found a correlation between TGF-β and IL-13, however, I don’t see that depicted in a graph. Furthermore, it’s very difficult to understand why BMI and cachexia are not correlated in this study.

Response 9. Figure 4 depicts the correlation of salivary TGF-β and IL-13 (only salivary, and that is also stated in the discussion). There is no correlation between serum concentrations of the cytokines, which is also depicted in Figure 4. The BMI and cancer cachexia are not correlated since the BMI level is predetermined in the cancer cachexia criteria. Therefore, the correlation is set, and since the cachexia is not graded, it provides no additional information. The BMI was included in the correlation analysis regarding the cytokine levels and logistic regression analysis. In logistic regression analysis it was clearly shown that BMI is statisticaly different in cachectic vs. non-cachectic patients (p<0,01 and p<0,001, respectively), what is shown in Table 1 and Table 3.

Reviewer 2 Report

Comments and Suggestions for Authors

Overall, the topic is significant and contributes to science.

1.       Materials and Methods: Please add the study design, including data collection time points.

2.       2.1. Patient selection and sampling (page 3)

a)       Inclusion/ exclusion criteria and how these patients were recruited are not clearly described. It is challenging to understand the type of patients included in the study, for example, cancer type, stages, treatment status, cancer status (newly diagnosed vs. late stage vs. in the middle of treatment for palliative intent or cure, etc.), or BMI as the inclusion criteria.

Exclusion criteria need more detailed information about how and when patients were excluded (lines 105-107 on page 3)

How many patients were screened?

How many patients withdrew from the study?

b)      Although the authors described how they divided patients into two groups (SCRINIO group criteria), it will be clear if they describe the two categories.

c)       Line 96 on page 3: The study was conducted from Feb. 2017 to June 2023. Please ensure the study period is correct because 16 years to collect data from 78 patients seems too long.

d)      Line 98 on page 3: A healthy control group includes 20 volunteers. There is no description of who would be considered healthy volunteers (inclusion/exclusion criteria), how they were recruited, and their roles in the study. No information about health control was provided in the results section.

e)      Lines 108-121 on page 4: Please provide information on when blood and saliva samples were collected regarding patient status (for example, a week after a diagnosis or right after recruitment, etc).

f)        Line 121 on page 4: “Weight and height were measured on the same day.” It is unclear “same day “ of recruitment, data collection, diagnosis, etc.

g)       Ethical Consideration: Please add the section on ethical consideration.

3.       Results (page 4)

a)       Table 1 does not include data on healthy volunteers. The total study sample should be 96 subjects instead of 76 since the healthy volunteers are the comparison group.

b)      Albumin: 40.6 g/L seems to be much higher than the reference range in the U.S. (35 g/L).

c)       Please add the legend by adding the complete spelling of the abbreviations in Tables 1, 2, and 3.

4.       Discussion: Please revise the discussion section to consider the inclusion/exclusion criteria and their demographic characteristics, including those of the control group.1.       Materials and Methods: Please add the study design, including data collection time points.

2.       2.1. Patient selection and sampling (page 3)

a)       Inclusion/ exclusion criteria and how these patients were recruited are not clearly described. It is challenging to understand the type of patients included in the study, for example, cancer type, stages, treatment status, cancer status (newly diagnosed vs. late stage vs. in the middle of treatment for palliative intent or cure, etc.), or BMI as the inclusion criteria.

Exclusion criteria need more detailed information about how and when patients were excluded (lines 105-107 on page 3)

How many patients were screened?

How many patients withdrew from the study?

b)      Although the authors described how they divided patients into two groups (SCRINIO group criteria), it will be clear if they describe the two categories.

c)       Line 96 on page 3: The study was conducted from Feb. 2017 to June 2023. Please ensure the study period is correct because 16 years to collect data from 78 patients seems too long.

d)      Line 98 on page 3: A healthy control group includes 20 volunteers. There is no description of who would be considered healthy volunteers (inclusion/exclusion criteria), how they were recruited, and their roles in the study. No information about health control was provided in the results section.

e)      Lines 108-121 on page 4: Please provide information on when blood and saliva samples were collected regarding patient status (for example, a week after a diagnosis or right after recruitment, etc).

f)        Line 121 on page 4: “Weight and height were measured on the same day.” It is unclear “same day “ of recruitment, data collection, diagnosis, etc.

g)       Ethical Consideration: Please add the section on ethical consideration.

3.       Results (page 4)

a)       Table 1 does not include data on healthy volunteers. The total study sample should be 96 subjects instead of 76 since the healthy volunteers are the comparison group.

b)      Albumin: 40.6 g/L seems to be much higher than the reference range in the U.S. (35 g/L).

c)       Please add the legend by adding the complete spelling of the abbreviations in Tables 1, 2, and 3.

4.       Discussion: Please revise the discussion section to consider the inclusion/exclusion criteria and their demographic characteristics, including those of the control group.

Comments on the Quality of English Language

The manuscript needs minor edits for flow and easy understanding.

Author Response

Thank you very much for your time and expert oppinion in reviewing this manuscript. Here are response on your questions/comments/suggestions, highlighted in turquoise in the text. 

Comments 1. Materials and Methods: Please add the study design, including data collection time points.

Response 1. Thank you for this important question which was not emphasized. The study design added and time points defined (line 104, 136, 137)

Comments 2. Patient selection and sampling (page 3)

a) Inclusion/ exclusion criteria and how these patients were recruited are not clearly described. It is challenging to understand the type of patients included in the study, for example, cancer type, stages, treatment status, cancer status (newly diagnosed vs. late stage vs. in the middle of treatment for palliative intent or cure, etc.), or BMI as the inclusion criteria.

Response 2a. The groups are described in more detail, and additional information about the study procedures is given in the text (line 112-137).

Comments 3. Exclusion criteria need more detailed information about how and when patients were excluded (lines 105-107 on page 3)

Response 3. Details are given in lines 121-130.

Question/comments 4. How many patients were screened?

Response 4. There were 82 patients eligible for sampling procedure and fullfilled criteria but in 4 of them we had no adequate quantity of sample (saliva)

 Comments/question 5. How many patients withdrew from the study?

- Response 5. None, study measurement was done once, no one withdrew patient consent

 Comments 6. Although the authors described how they divided patients into two groups (SCRINIO group criteria), it will be clear if they describe the two categories.

Response 6.The categories are described in line 112-118

Comments 7. Line 96 on page 3: The study was conducted from Feb. 2017 to June 2023. Please ensure the study period is correct because 16 years to collect data from 78 patients seems too long.

Response 7. The timeline between 2017 and 2023 is six years and an accurate timeline of the study.

Comments 8.  Line 98 on page 3: A healthy control group includes 20 volunteers. There is no description of who would be considered healthy volunteers (inclusion/exclusion criteria), how they were recruited, and their roles in the study. No information about health control was provided in the results section.

Response 8. Additional information about healthy controls added to line 130 + line 175-176.

 Question/comments 9. Lines 108-121 on page 4: Please provide information on when blood and saliva samples were collected regarding patient status (for example, a week after a diagnosis or right after recruitment, etc).

Response 9. Details are provided as requested (line 136-137)

Comments/question 10.  Line 121 on page 4: “Weight and height were measured on the same day.” It is unclear “same day “ of recruitment, data collection, diagnosis, etc.

 Response 10. details are provided in the text (line 146).

Comments/suggestion 11. Ethical Consideration: Please add the section on ethical consideration.

Respons 11. Institutional board review and ethical considerations are given at the end of the manuscript before the references (additional information about ethical issues added) as per the paper's instructions for authors (line 379-381)

Comments 12. Results (page 4)

  1. a) Table 1 does not include data on healthy volunteers. The total study sample should be 96 subjects instead of 76 since the healthy volunteers are the comparison group.

Response 12. Thank you for this important question. We agree it was not put in the table. Content of Table 1 is based on patients' characteristics, only (all together 78 patients) and additional information about healthy controls is added in the text (line 175-177).

Commentes 13. b) Albumin: 40.6 g/L seems to be much higher than the reference range in the U.S. (35 g/L).

Response 13.  The reference values for albumin concentrations were defined by the local laboratory (all the subjects were tested there), accredited by local and international agencies.

Comments 14. c) Please add the legend by adding the complete spelling of the abbreviations in Tables 1, 2, and 3.

Response 14. Legends added as it was required.

Comments 15. Discussion: Please revise the discussion section to consider the inclusion/exclusion criteria and their demographic characteristics, including those of the control group.

Response 15. Additional information added (line 366-371).

Round 2

Reviewer 2 Report

Comments and Suggestions for Authors

Dear Authors,

Please see the further comments to your responses in red font, which suggests more revisions or clarifications.

***************************************************************

Thank you very much for your time and expert opinion in reviewing this manuscript. Here are response on your questions/comments/suggestions, highlighted in turquoise in the text. 

Thank you for being considerate in highlighting the revised sections for easy identification. Much appreciated,

Comments 1. Materials ad Methods: Please add the study design, including data collection time points.

Response 1. Thank you for this important question which was not emphasized. The study design added and time points defined (line 104, 136, 137)_

Q1. Thank you for adding the study design (cross-sectional observational study). The biomarker samples were collected one time point only; however, the study participants were in different phases of their cancer treatment (during treatment and follow-up- line 106, page3). If so, their biomarker results may differ between patients in active treatment and those in follow-up after completion of treatment). I would suggest the subgroup analysis to check the differences between active treatment group and follow-up group. Otherwise, please clarify what it means by ‘during treatment and follow-up.’Line 107 on page 3. Please describe how the healthy volunteers were recruited and the definition of health volunteers for this study.

Comments 2. Patient selection and sampling (page 3)

a) Inclusion/ exclusion criteria and how these patients were recruited are not clearly described. It is challenging to understand the type of patients included in the study, for example, cancer type, stages, treatment status, cancer status (newly diagnosed vs. late stage vs. in the middle of treatment for palliative intent or cure, etc.), or BMI as the inclusion criteria.

Response 2a. The groups are described in more detail, and additional information about the study procedures is given in the text (line 112-137).

Q2a-1. 10% of body weight loss in how many months? For example, weight loss in 3 months, 6 months, or 1 months?

Q2a-2. Line 114  on page 4. Please check the criteria for the persons with” a BMI of less than 20 kg/m2”.There are different requirements to be defined as ‘cachectic’ between BMI greater than 20 kg/m2 and less than 20 kg/m2. Please revisit your original participants with BMI less than 20 kg/m2 to make sure they lost the weight in X months (usually 1 month) and lose X% of weight to be defined as cachectic. Do you define a patient to be cachectic if the BMI is less than 20kg/m2 although this/her weight has been stable for a long time without weight loss? Line 117 on page 4. I suggest removing ‘best’ in front of supportive care (line 118)so the sentence can be more objective.

Lines 136-137.Please clarify  “conducted one day after recruitment and after at least one radiological reevaluation during the cancer treatment.” It is not clear if the research team collected samples both times (a day after recruitment and after at least one radiological reevaluation based on the earlier statement about cross-setional study with one sampling point.

Comments 3. Exclusion criteria need more detailed information about how and when patients were excluded (lines 105-107 on page 3)

Response 3. Details are given in lines 121-130.

Question/comments 4. How many patients were screened?

Response 4. There were 82 patients eligible for sampling procedure and fullfilled criteria but in 4 of them we had no adequate quantity of sample (saliva) _Please reflect this information in your manuscript.

 Comments/question 5. How many patients withdrew from the study?

- Response 5. None, study measurement was done once, no one withdrew patient consent Please reflect this information in your manuscript.

 Comments 6. Although the authors described how they divided patients into two groups (SCRINIO group criteria), it will be clear if they describe the two categories.

Response 6.The categories are described in line 112-118

Comments 7. Line 96 on page 3: The study was conducted from Feb. 2017 to June 2023. Please ensure the study period is correct because 16 years to collect data from 78 patients seems too long.

Response 7. The timeline between 2017 and 2023 is six years and an accurate timeline of the study.

Comments 8.  Line 98 on page 3: A healthy control group includes 20 volunteers. There is no description of who would be considered healthy volunteers (inclusion/exclusion criteria), how they were recruited, and their roles in the study. No information about health control was provided in the results section.

Response 8. Additional information about healthy controls added to line 130 + line 175-176. The information provided in the result section is not sufficient for the comment. Percentages of healthy volunteers are addressed. What was the purpose of using the volunteers? If the purpose of having the healthy controls to compare tthem to cancer patients with/without cachexia, add their results in Table 1.

 Question/comments 9. Lines 108-121 on page 4: Please provide information on when blood and saliva samples were collected regarding patient status (for example, a week after a diagnosis or right after recruitment, etc).

Response 9. Details are provided as requested (line 136-137)

Comments/question 10.  Line 121 on page 4: “Weight and height were measured on the same day.” It is unclear “same day “ of recruitment, data collection, diagnosis, etc.

 Response 10. details are provided in the text (line 146).

Comments/suggestion 11. Ethical Consideration: Please add the section on ethical consideration.

Respons 11. Institutional board review and ethical considerations are given at the end of the manuscript before the references (additional information about ethical issues added) as per the paper's instructions for authors (line 379-381)

Comments 12. Results (page 4)

  1. a) Table 1 does not include data on healthy volunteers. The total study sample should be 96 subjects instead of 76 since the healthy volunteers are the comparison group.

Response 12. Thank you for this important question. We agree it was not put in the table. Content of Table 1 is based on patients' characteristics, only (all together 78 patients) and additional information about healthy controls is added in the text (line 175-177). It is not clear to have a group of healthy volunteers if their data are not going to be used at all. Descriptions of how many men and women in the healthy volunteer group is not enough to benefit the study and might not be the purpose of having the volunteers in the study. Please add their lab to compare to the cancer patients in Table 1.

Comments 13. b) Albumin: 40.6 g/L seems to be much higher than the reference range in the U.S. (35 g/L).

Response 13.  The reference values for albumin concentrations were defined by the local laboratory (all the subjects were tested there), accredited by local and international agencies.

Comments 14. c) Please add the legend by adding the complete spelling of the abbreviations in Tables 1, 2, and 3.

Response 14. Legends added as it was required._Thank you

Comments 15. Discussion: Please revise the discussion section to consider the inclusion/exclusion criteria and their demographic characteristics, including those of the control group.

Response 15. Additional information added (line 366-371). My apologies if my intention for this comment was not clear. I did not mean that the research team needed to add the exclusion criteria in the Discussion section (it should be included in the patient selection and sample section). Inclusion/exclusion criteria may affect the interpretation of the findings and should be reflected in the Discussion section.

Author Response

Dear Sirs,

thank you for your time and expertise in analysing our manuscript. Here are the response to your questions/suggestions. Additional changes in the text are highlighted in purple, to be easier to find. 

Comments 1. Materials ad Methods: Please add the study design, including data collection time points.

Response 1. Thank you for this important question which was not emphasized. The study design added and time points defined (line 104, 136, 137)_

Q1. Thank you for adding the study design (cross-sectional observational study). The biomarker samples were collected one time point only; however, the study participants were in different phases of their cancer treatment (during treatment and follow-up- line 106, page3). If so, their biomarker results may differ between patients in active treatment and those in follow-up after completion of treatment). I would suggest the subgroup analysis to check the differences between active treatment group and follow-up group. Otherwise, please clarify what it means by ‘during treatment and follow-up.’Line 107 on page 3. Please describe how the healthy volunteers were recruited and the definition of health volunteers for this study.

Answer on Q1. We think that cancer treatment was not essential in the process of cachexia development, although we can accept that it might have some importance. Thus, the idea was that at some point, during treatment AND follow-up, biomarkers appear in saliva or blood, as a reflection of metabolic changes in direction of cachexia. Thank you for your suggestion, which is interesting, to check differences in subgroup, but our design was not going in that way. After all, the subgroups would be too small for statistic analysis. The meaning of follow-up is that some of patients were not all the time in active treatment. As it is very well known, in clinical oncology some patients have pause or stop of treatment for different reasons. We didn't want this real-life situation to be limitation for the purpose of our research. Just to stress again, that all patients were in active treatment as it was clearly stated in manuscript (see line 116-117).  In some future analysis, it is, of course, justified to strictly define groups of acitve treatment patients and patients without active treatment. Regarding healthy volunteers – these are individuals with no known malignant tumor or any other chronic disease. Usually we asked professional stuff from hospital to join (nurses, doctors etc)   

Comments 2. Patient selection and sampling (page 3)

  1. a) Inclusion/ exclusion criteria and how these patients were recruited are not clearly described. It is challenging to understand the type of patients included in the study, for example, cancer type, stages, treatmentstatus, cancer status (newly diagnosed vs. late stage vs. in the middle of treatment for palliative intent or cure, etc.), or BMI as the inclusion criteria.

Response 2a. The groups are described in more detail, and additional information about the study procedures is given in the text (line 112-137).

Q2a-1. 10% of body weight loss in how many months? For example, weight loss in 3 months, 6 months, or 1 months?

Answer Q2a-1. 10% body weight loss in 3 months: Please, see ref.18 in literature. We didn't emphasize this time frameshift, it can be put in the text. We found it not crucial, but of course it is logic question, thank you.

Q2a-2Line 114  on page 4. Please check the criteria for the persons with” a BMI of less than 20 kg/m2”.There are different requirements to be defined as ‘cachectic’ between BMI greater than 20 kg/m2 and less than 20 kg/m2. Please revisit your original participants with BMI less than 20 kg/m2 to make sure they lost the weight in X months (usually 1 month) and lose X% of weight to be defined as cachectic. Do you define a patient to be cachectic if the BMI is less than 20kg/m2 although this/her weight has been stable for a long time without weight loss? Line 117 on page 4. I suggest removing ‘best’ in front of supportive care (line 118)so the sentence can be more objective.

Answer Q2a-2. As it was said, we used SCRINIO working group criteria (line 111-114), one of the reasons was its simplicity and applicability. For our aim, it was easy-to-use, although there are different criteria that can be used. It is not „just“ BMI less than 20 kg/m2.

„Best“ is removed from „best supportive care“, line 118. It is usual idiom in english for patients not treated with specific systemic treatment

Lines 136-137.Please clarify  “conducted one day after recruitment and after at least one radiological reevaluation during the cancer treatment.” It is not clear if the research team collected samples both times (a day after recruitment and after at least one radiological reevaluation based on the earlier statement about cross-setional study with one sampling point.

Answer: text modified – the sampling was conducted one day after the patients gave their consent and inclusion and exclusion criteria were checked, and a phrase that the sampling was conducted after at least one radiological evaluation describes the timing of the sampling in the course of the disease (only one sampling overall) – this phrase was moved to line 117 (highlighted in purple) to clear the non-intentional unclarity which may have occurred.

Comments 3. Exclusion criteria need more detailed information about how and when patients were excluded (lines 105-107 on page 3)

Response 3. Details are given in lines 121-130.

Question/comments 4. How many patients were screened?

Response 4. There were 82 patients eligible for sampling procedure and fullfilled criteria but in 4 of them we had no adequate quantity of sample (saliva) _Please reflect this information in your manuscript.

Text added as instructed (highlighted in purple) line 137-139.

 Comments/question 5. How many patients withdrew from the study?

- Response 5. None, study measurement was done once, no one withdrew patient consent Please reflect this information in your manuscript.

Text added as instructed (highlighted in purple) line 139.

 Comments 6. Although the authors described how they divided patients into two groups (SCRINIO group criteria), it will be clear if they describe the two categories.

Response 6.The categories are described in line 112-118

Comments 7. Line 96 on page 3: The study was conducted from Feb. 2017 to June 2023. Please ensure the study period is correct because 16 years to collect data from 78 patients seems too long.

Response 7. The timeline between 2017 and 2023 is six years and an accurate timeline of the study.

Comments 8.  Line 98 on page 3: A healthy control group includes 20 volunteers. There is no description of who would be considered healthy volunteers (inclusion/exclusion criteria), how they were recruited, and their roles in the study. No information about health control was provided in the results section.

Response 8. Additional information about healthy controls added to line 130 + line 175-176. The information provided in the result section is not sufficient for the comment. Percentages of healthy volunteers are addressed. What was the purpose of using the volunteers? If the purpose of having the healthy controls to compare tthem to cancer patients with/without cachexia, add their results in Table 1.

Additional explanation on Response 8. The purpose of using volunteers was to check the difference between healthy individuals and cancer patients. Data were not shown in the table, it was more internal information and proof of being on the right direction. The central question was cachectic vs non-cachectic – what makes difference between these groups. We didn't want to lose the point of this analysis. Additionaly, there was no routine biochemical analysis, only level of cytokines, for the healthy volunteer group. We think it was not relevant for the problem analysed in this paper.

 Question/comments 9. Lines 108-121 on page 4: Please provide information on when blood and saliva samples were collected regarding patient status (for example, a week after a diagnosis or right after recruitment, etc).

Response 9. Details are provided as requested (line 136-137)

Comments/question 10.  Line 121 on page 4: “Weight and height were measured on the same day.” It is unclear “same day “ of recruitment, data collection, diagnosis, etc.

 Response 10. details are provided in the text (line 146).

Comments/suggestion 11. Ethical Consideration: Please add the section on ethical consideration.

Respons 11. Institutional board review and ethical considerations are given at the end of the manuscript before the references (additional information about ethical issues added) as per the paper's instructions for authors (line 379-381)

Comments 12. Results (page 4)

  1. a) Table 1 does not include data on healthy volunteers. The total study sample should be 96 subjects instead of 76 since the healthy volunteers are the comparison group.

Response 12. Thank you for this important question. We agree it was not put in the table. Content of Table 1 is based on patients' characteristics, only (all together 78 patients) and additional information about healthy controls is added in the text (line 175-177). It is not clear to have a group of healthy volunteers if their data are not going to be used at all. Descriptions of how many men and women in the healthy volunteer group is not enough to benefit the study and might not be the purpose of having the volunteers in the study. Please add their lab to compare to the cancer patients in Table 1.

It was already explained (see above – Additional explanation on Response 8) – in the rather small group of healthy controls we didn't analyse gender structure (or other characteristics) of the group. It was not the idea of this analysis, it was also not the point neither is relevant for research purpose. Please, see earlier explanation regarding this issue.

Comments 13. b) Albumin: 40.6 g/L seems to be much higher than the reference range in the U.S. (35 g/L).

Response 13.  The reference values for albumin concentrations were defined by the local laboratory (all the subjects were tested there), accredited by local and international agencies.

Comments 14. c) Please add the legend by adding the complete spelling of the abbreviations in Tables 1, 2, and 3.

Response 14. Legends added as it was required._Thank you

Comments 15. Discussion: Please revise the discussion section to consider the inclusion/exclusion criteria and their demographic characteristics, including those of the control group.

Response 15. Additional information added (line 366-371). My apologies if my intention for this comment was not clear. I did not mean that the research team needed to add the exclusion criteria in the Discussion section (it should be included in the patient selection and sample section). Inclusion/exclusion criteria may affect the interpretation of the findings and should be reflected in the Discussion section.

Response 15. Thank you again for this question. It was already explained in previous review, so it was added in the Discussion section as you asked. Once again, by exclusion criteria we tried to eliminate possible conditions that lead to increased levels of cytokines of our interest. It is probably not possible to exclude all the potential influences but we were not aware of that while doing this analysis.